# Liposomal Nanovaccine Containing α-Galactosylceramide and Ganglioside GM3 Stimulates Robust CD8^+^ T Cell Responses via CD169^+^ Macrophages and cDC1

**DOI:** 10.3390/vaccines9010056

**Published:** 2021-01-16

**Authors:** Joanna Grabowska, Dorian A. Stolk, Maarten K. Nijen Twilhaar, Martino Ambrosini, Gert Storm, Hans J. van der Vliet, Tanja D. de Gruijl, Yvette van Kooyk, Joke M.M. den Haan

**Affiliations:** 1Department of Molecular Cell Biology and Immunology, Amsterdam UMC, Cancer Center Amsterdam, Amsterdam Infection and Immunity Institute, Vrije Universiteit Amsterdam, 1081 HZ Amsterdam, The Netherlands; j.grabowska@amsterdamumc.nl (J.G.); d.stolk@amsterdamumc.nl (D.A.S.); m.nijentwilhaar@amsterdamumc.nl (M.K.N.T.); ambrosinimartino78@gmail.com (M.A.); y.vankooyk@amsterdamumc.nl (Y.v.K.); 2Department of Pharmaceutics, Utrecht Institute for Pharmaceutical Sciences, Utrecht University, 3584 CG Utrecht, The Netherlands; G.Storm@uu.nl; 3Department of Biomaterials Science and Technology, University of Twente, 7500 AE Enschede, The Netherlands; 4Department of Surgery, Yong Loo Lin School of Medicine, National University of Singapore, Singapore 119228, Singapore; 5Department of Medical Oncology, Amsterdam UMC, Cancer Center Amsterdam, Amsterdam Infection and Immunity Institute, Vrije Universiteit Amsterdam, 1081 HV Amsterdam, The Netherlands; jj.vandervliet@amsterdamumc.nl (H.J.v.d.V.); td.degruijl@amsterdamumc.nl (T.D.d.G.); 6Lava Therapeutics, 3584 CM Utrecht, The Netherlands

**Keywords:** vaccination, liposomes, anti-tumor, CD169 macrophage, cDC1, invariant natural killer T cell, alpha galactosylceramide, ganglioside GM3

## Abstract

Successful anti-cancer vaccines aim to prime and reinvigorate cytotoxic T cells and should therefore comprise a potent antigen and adjuvant. Antigen targeting to splenic CD169^+^ macrophages was shown to induce robust CD8^+^ T cell responses via antigen transfer to cDC1. Interestingly, CD169^+^ macrophages can also activate type I natural killer T-cells (NKT). NKT activation via ligands such as α-galactosylceramide (αGC) serve as natural adjuvants through dendritic cell activation. Here, we incorporated ganglioside GM3 and αGC in ovalbumin (OVA) protein-containing liposomes to achieve both CD169^+^ targeting and superior DC activation. The systemic delivery of GM3-αGC-OVA liposomes resulted in specific uptake by splenic CD169^+^ macrophages, stimulated strong IFNγ production by NKT and NK cells and coincided with the maturation of cDC1 and significant IL-12 production. Strikingly, superior induction of OVA-specific CD8^+^ T cells was detected after immunization with GM3-αGC-OVA liposomes. CD8^+^ T cell activation, but not B cell activation, was dependent on CD169^+^ macrophages and cDC1, while activation of NKT and NK cells were partially mediated by cDC1. In summary, GM3-αGC antigen-containing liposomes are a potent vaccination platform that promotes the interaction between different immune cell populations, resulting in strong adaptive immunity and therefore emerge as a promising anti-cancer vaccination strategy.

## 1. Introduction

For many decades the combination of chemotherapy and surgery has been the standard of care for a variety of cancer types. The rise of immunotherapy, e.g., immune checkpoint inhibition (ICI) that targets PD-1 or CTLA-4, has brought an apparent shift towards immune-related treatments. Although ICI as stand-alone therapy has shown remarkable response rates of up to 52% in different cohorts of patients with solid malignancies such as melanoma, renal carcinoma or non-small cell lung cancer, a substantial proportion of the patients are partially- or non-responsive to ICI-based monotherapy [1,2]. Considerable efforts have been made to investigate the combination of ICI with other anti-cancer interventions. Since ICI counteracts dysfunction of tumor-reactive T cells and since poorly immunogenic tumors lack sufficient infiltration by T cells (TIL), there is a strong rationale for combining ICI with vaccination strategies [3,4].

Therapeutic vaccination strategies aim to effectively induce CD8^+^ T cell activation towards well-characterized tumor antigens. Dendritic cells (DC), as professional antigen presenting cells, exhibit superior T cell (cross)-priming capacity. In particular cDC1 cells, which are characterized by expression of XCR1 and are dependent on the transcription factor Batf3, exert a key role in the induction of CD8^+^ T cells and have been shown to be essential for the rejection of highly immunogenic tumors and for the efficacy of ICI in mouse models [5,6,7,8]. For this reason, many DC-based immunotherapies have been investigated, with the main goal being to induce anti-tumor responses. Interestingly, macrophages have also been proposed to actively contribute in the priming of T cells via collaboration with, and antigen transfer to, cDC1 [9,10,11], in particular the macrophages expressing CD169, also known as Siglec-1, that are located in the marginal zone (MZ) of the spleen (metallophilic MZ macrophages) and in the subcapsular sinus (SCS) of the lymph node (LN). CD169^+^ macrophages have been reported to play a protective role against bacterial and viral infections and therefore have been termed gatekeepers [12], but their role in anti-tumor responses has also been described. Targeting this particular macrophage subset with a monoclonal antibody against CD169 coupled to ovalbumin (OVA) in the presence of adjuvant generated strong cytotoxic T cell (CTL) responses, which were absent after macrophage depletion using clodronate liposomes [9,11], and delayed tumor growth in the B16-OVA melanoma mouse model [13]. A high density of CD169^+^ macrophages in the LN sinus was shown to be a predictive factor for the clinical prognosis in cancer patients [14]. Moreover, liposomal strategies exploiting a synthetic CD169 ligand showed very specific binding to CD169^+^ bone marrow-derived macrophages and induced the proliferation of OVA-specific T cells [15]. Similarly, liposomes with a non-synthetic i.e., endogenous ligand for CD169, the sialic acid-containing glycosphingolipid GM3 ganglioside [16], have been reported to effectively bind to mouse CD169^+^ macrophages in the spleen and to human monocyte-derived and primary DCs and stimulate T cell responses [17,18,19]. Together, these data illustrate that ligand targeting in addition to antibody targeting is an effective strategy for antigen delivery to CD169^+^ macrophages. 

Besides their role in antigen uptake and antigen transfer to cDC1, CD169^+^ macrophages have also been proven to be successful activators of invariant (Type I) natural killer T cells (NKT) in both the spleen and the LN after delivery of the glycolipid α-galactosylceramide (αGC) [20,21,22]. Since effective therapeutic vaccination not only relies on the optimal induction of adaptive immune responses, but also on strong stimulation of innate players of the immune system, the inclusion of an NKT-activating ligand presents a good rationale for vaccine optimization. αGC is a potent anti-tumor agent, unleashing the cytotoxic potential of NKT and causing the subsequent activation of other immune cells, such as natural killer cells (NK) cells, B cells and DCs [23,24,25]. αGC-activated NKT produce IFNy, induce IFNy secretion from NK cells [26], and further stimulate DCs for the activation of CD8^+^ T cells [27]. While no clinical benefit was observed when αGC was used as a single modality (injected intravenously [i.v.]), NKT expansion and immune activation was reported in patients treated with αGC-pulsed DCs [28,29]. On the other hand, multiple pre-clinical studies using αGC-containing vaccination strategies have been shown to effectively inhibit tumor growth. As such, tumor reduction in a liver metastasis model of B16 melanoma was observed and was associated with enhanced immune infiltration by CD8^+^ T cells, macrophages and NK cells [30]. This anti-tumor potential of NKT-adjuvanted vaccines containing αGC was further demonstrated in Hepatis B virus-related hepatocellular carcinoma [31], lymphoma [32] and multiple B16 mouse model studies in which αGC in combination with antigen was directly targeted to DCs [33,34,35].

We hypothesized that combining a strong CD169 ligand and a potent NKT cell activator in one vaccine formulation could act synergistically to reinforce the anti-tumor immune response. We developed a single liposome-based vaccination modality which was composed of ganglioside GM3 for specific targeting to CD169^+^ macrophages and the NKT activating ligand αGC, both incorporated in the liposomal bilayer, and OVA, as a model antigen, that was encapsulated in the aqueous core. With this liposomal formulation we demonstrate that the incorporation of both GM3 and αGC augments the CD8^+^ T cell activating capacity of OVA-containing liposomes and evokes B-cell activation. We show that superior antigen-specific T cell immunity is dependent on cDC1 and CD169^+^ macrophages and associated with strong NKT and NK cell activation. Therefore, GM3- and αGC-containing liposomes exhibit excellent potential as a vaccination platform that promotes interplay between CD169^+^ macrophages, cDC1 and NKT, ultimately mediating robust antigen-specific immune responses.

## 2. Materials and Methods 

### 2.1. Animals

C57BL/6 WT and Batf3KO mice were bred in-house in the animal facility of Amsterdam UMC (location VUmc), Amsterdam, The Netherlands. CD169 mutant animals (W2QR97A) were generated at the University of Dundee and were a kind gift from Prof. P.R. Crocker (Dundee, Scotland) [36]. CD169- diphtheria toxin receptor (DTR) mice were generated by Dr. K. Asano and Dr. M. Tanaka from the Tokyo University of Pharmacy and Life Sciences (Tokyo, Japan) and were a kind gift from Dr. A. Hidalgo (Madrid, Spain) [37,38]. Males and females between 10 and 20 weeks were used for experimental procedures. The animal work was performed in accordance with Dutch government guidelines and approved by Animal Experiment Committee (DEC) and Central Committee on Animal Experiments (CCD, ADV1140020171024).

### 2.2. Liposome Preparation

OVA-containing GM3, αGC and GM3-αGC liposomes were prepared using the dry film extrusion technique at the Department of Pharmaceutics, Faculty of Science, Utrecht University (Utrecht, The Netherlands), as previously described [39]. In brief, a lipid mixture containing egg phosphatidylcholine (EPC)-35 (Lipoid), egg phosphatidylglycerol (EPG)-Na (Lipoid) and Cholesterol (Sigma-Aldrich, Darmstadt, Germany) (molar ratio 3.8:1:2.5) dissolved in chloroform/methanol (2:1) was combined with 3 mol% GM3 ganglioside and/or α-galactosylceramide, KRN7000 (Funakoshi, Tokyo, Japan) (30 μg) and 0.1 mol% of the lipophilic fluorescent tracer DiD (1′-dioctadecyl-3,3,3′,3′-tetramethyl indodicarbocyanine, Life Technologies, Frederick, MD, USA). Following organic solvent evaporation in a rotary evaporator, the obtained lipid film was rehydrated in 1 mg/mL Ovalbumin solution (OVA, Calbiochem, San Diego, CA, USA). In order to obtain nanoparticles about 200 nm in size, the solution was extruded five times through stacked 400 nm and 200 nm filters using a high-pressure extruder. The non-encapsulated OVA protein was removed by two sequential ultracentrifugation steps. Pelleted liposomes were resuspended in HEPES buffer (pH 7.5) containing antibiotics (50 U/mL penicillin and 50 μg/mL streptomycin, Lonza, Basel, Switzerland) by vigorous vortexing. Physiochemical characterization of the obtained nanoparticles was performed using dynamic light scattering (DLS) using a Zetasizer Nano ZSP (Malvern Instruments, Malvern, UK), which provides the mean size, polydispersity index, and zeta potential. Antigen encapsulation efficiency was around 2% (15–20 µg/mL) as determined by sandwich ELISA. The amount of αGC in the liposomes was determined as described previously (+/−20 ug/mL) [39]. The liposome dose used in the experiments was based on the molar concentration of phospholipids determined as described previously [40]. The liposomes were stored at 4 °C and used within 2 months after preparation.

### 2.3. CD169-Fc ELISA

25µmoles of liposomes were dissolved in absolute ethanol and coated onto MaxiSorp ELISA plates (NUNC, Roskilde, Denmark) and left to air dry overnight. The plates were incubated with 1% BSA/PBS for 1 h at RT. Next, 2 µg/mL mouse CD169-Fc conjugate WT or mutant (mouse CD169 fused to Fc fragment of human IgG1) was added for 1 h at RT [36]. Next, the plates were incubated with peroxidase (PO) goat anti-human IgG-Fc in 1% BSA/PBS for 30 min at RT. Following a washing step with PBS, the reaction was developed with 100 μg/mL of TMB (Sigma-Aldrich, Darmstadt, Germany) and 0.006% hydrogen peroxide in substrate buffer. The absorbance was measured at 450 nm using a microplate absorbance spectrophotometer (Biorad, Hercules, CA, USA).

### 2.4. In Vivo Immunization, Spleen Digestion and Re-Stimulation

Mice were injected i.v. in the tail vein with 30 nM of liposomal solution (containing 200 ng αGC and 150 ng OVA) in PBS on day 0 (0 h) and splenocytes were collected 2 h post injection (p.i)., 16 h p.i. or 7 days p.i. Where indicated, GM3-OVA liposomes were co-injected with 25 µg αCD40 (in-house made) and 25 µg polyI:C (Invivogen, Toulouse, France). Spleens collected at 2 h p.i. and 16 h p.i. were first mechanistically dissociated, then exposed to enzymatic digestion with 4 mg/mL Lidocaine, 2 WU/mL Liberase TL (Roche, Mannheim, Germany) and 50 µg/mL DNase I (Roche, Mannheim, Germany) and finally treated with ammonium-chloride-potassium lysis buffer to lyse red blood cells. The obtained cell suspensions were filtered through 100 µm filter and stained with surface antibody mixture to be further analyzed by flow cytometry. To investigate immune responses in liposome-immunized mice, spleens collected on day 7 p.i. were mashed, filtered through 70 µm filter, and treated with ammonium-chloride-potassium lysis buffer. The obtained cell suspensions were either directly stained with surface antibody mixture for further flow cytometry analysis or re-stimulated with OVA peptides. The re-stimulation protocol involved incubation with the MHC I-restricted OVA 257–264 peptide (0.1 µg/mL) or the MHC II-restricted OVA 262–276 peptide (100 µg/mL) for 5 h or 24 h, respectively, in the presence of GolgiPlug during the last 5 h (BD Biosciences, San Jose, CA, USA).

### 2.5. Flow Cytometry 

Single-cell suspensions were first incubated with FcR block (2.4G2 clone, in-house made) for 15 min on ice and next stained with a surface antibody mix for 30 min on ice (Table 1). For NKT staining, after FcR block, the cells were incubated with mouse CD1d PBS-57 tetramer for 30 min at RT (NIH Tetramer Core Facility, Atlanta, GA, USA) and then with a surface antibody mix for 30 min on ice. Next, the cells were fixed using 2% paraformaldehyde and acquired on Fortessa (BD Biosciences, San Jose, CA, USA). For intracellular staining, following fixation, the cells were washed 2 times with 0.5% Saponin solution to permeabilize the membrane and then incubated with an intracellular antibody mix diluted in 0.5% saponin. Flow cytometry data were analyzed using FlowJo V10 software.

### 2.6. Detection of Anti-OVA Ig in the Serum

Blood was collected from liposome-immunized mice at day 7 p.i. via heart puncture and centrifuged to obtain serum. MaxiSorp ELISA plates (NUNC, Roskilde, Denmark) were coated with 5 μg/mL OVA (Sigma-Aldrich, Darmstadt, Germany) in sodium phosphate buffer (Na_2_HPO_4_, NaH_2_PO_4_ and MiliQ, pH 6.5) o/n at 4 °C. After washing with 0.05% Tween20/PBS, a blocking step with 1% BSA/PBS for 1 h at RT was performed and the plates were next incubated with serial dilutions of serum in 1% BSA/PBS for 2 h at RT. Following a subsequent washing step, rabbit anti-mouse Ig-HRP (Dako, Santa Clara, CA, USA) was added for 1 h at RT. After washing, the reaction was developed with 100 μg/mL of TMB (Sigma-Aldrich, Darmstadt, Germany) and 0.006% hydrogen peroxide in substrate buffer. The absorbance was measured at 450 nm using a microplate absorbance spectrophotometer (Biorad, Hercules, CA, USA). The cut-off value was calculated as the mean of the control wells (no serum) plus 3x SD. Serum dilutions with OD values higher than or equal to the cut-off value were determined as antibody titer.

### 2.7. Detection of Cytokines in the Serum

Blood was collected from liposome-immunized C57BL/6 WT mice at 16 h p.i. via heart puncture and centrifuged to obtain serum. To quantify multiple cytokines in the serum, a bead-based immunoassay LEGENDplex (Biolegend, San Diego, CA, USA) was performed according to manufacturer’s instructions. Briefly, diluted serum and standard were mixed with beads and assay buffer and incubated for 2 h at RT. After a washing procedure, detection antibodies were added for 1 h at RT and the samples were next incubated with Streptavidin-PE for 30 min at RT, then washed and acquired on Fortessa (BD Biosciences, San Jose, CA, USA). An online software tool provided by the manufacturer was used to analyze the data.

### 2.8. Statistical Analysis

Statistical analysis of the obtained data was performed using GraphPad Prism 8. A one-way ANOVA test with Tukey’s multiple comparison test or Kruskal–Wallis test were performed to determine statistical significance. Where *p* values were near-significant, exact values were provided.

## 3. Results

### 3.1. Incorporation of GM3 in Liposomes Results in Increased Uptake by Splenic CD169^+^ Macrophages

The liposomes used in this study consisted of a phospholipid bilayer made of cholesterol, EPC-35 and EPG-Na that surrounded the aqueous core encapsulating the OVA protein. Additionally, ganglioside GM3 for specific CD169 targeting and/or glycolipid αGC, as NKT activating ligand, were incorporated into the bilayer to generate 3 different liposomes: GM3-OVA, αGC-OVA and GM3-αGC-OVA (Figure 1A). DiD, a fluorescent lipophilic dye, was integrated into the liposomal membrane to enable nanoparticle tracking. We designed negatively charged nanoparticles that were on average between 150–200 nm in size (Figure 1B). The presence of the GM3 molecule, a ligand that binds the CD169 receptor via a terminal positioned α2,3-linked sialic acid [41], in liposomes was confirmed with an ELISA using CD169-Fc, that revealed binding of GM3 to WT- but not the mutant form of CD169 (Figure 1C). To investigate the targeting potential of GM3-containing liposomes, WT mice and CD169 mutant mice, which have a non-functional CD169 receptor due to two mutations (W2QR97A), limiting ligand binding [36], were injected i.v. with different liposomal formulations. Two hours post injection (p.i.) the DiD signal associated with splenic CD169^+^ macrophages was assessed as a measurement for liposomal capture (Appendix A for gating strategy). While all liposomes appeared to be captured by CD169^+^ macrophages, the modification of liposomes with GM3 led to a significantly higher nanoparticle uptake, reflected by the increased geometric mean fluorescent intensity (gMFI) of the DiD signal, compared to the αGC-OVA liposomes. The observed effect was completely abrogated in CD169 mutant mice (Figure 1D), demonstrating an indisputable role of the CD169 receptor in the augmented uptake. In contrast to CD169^+^ macrophages, red pulp macrophages sequestered nanoparticles to a considerably lesser extent, and furthermore only marginal uptake by cDC1 and B cells was detected (Appendix A). In conclusion, the incorporation of GM3 into the bilayer of OVA-containing liposomes greatly enhances liposome uptake by CD169^+^ macrophages in vivo and is therefore an efficient targeting tool for the enhanced specific delivery of antigen- and αGC-containing nanoparticles.

### 3.2. Combining αGC and GM3 in OVA-Containing Liposomes Results in Potent NKT and NK Activation and Generates Robust Antigen-Specific CD8^+^ T Cell Responses

After observing an increased uptake of GM3-modified OVA-containing liposomes by CD169^+^ macrophages, we investigated the effect of αGC inclusion on NKT activation and measured IFNγ production and expression levels of activation markers 2 h p.i. Although the frequency of NKT at this timepoint did not vary between mice that received different liposomal formulations (data not shown), the incorporation of αGC into liposomes led to IFNγ production by 80% of NKT, which was accompanied by upregulation of the activation markers CD69 and CD25 (Figure 2A). Surprisingly, regardless of the presence of a CD169 targeting moiety, both αGC-containing liposomes were equally capable of stimulating NKT, as reflected by similar levels of IFNγ secretion. Notably, administration of αGC-containing liposomes contributed to potent NK cell responses, as illustrated by the frequency of IFNγ-producing cells (Figure 2B). Similarly, inclusion of αGC in the OVA and GM3-OVA liposomes was equally effective in stimulating NK cells. Taken together, these data indicate that nanoparticles containing αGC exhibit great capacity to induce NKT and NK cell activation.

Next, we evaluated the effect of combining GM3 and αGC in OVA-containing liposomes on the adaptive immune responses, by analyzing spleen 7 days p.i. Immunization with GM3-αGC-OVA nanoparticles resulted in the highest frequency of antigen specific CD8^+^ T cells and thus was superior to immunization with other liposomal formulations (Figure 2C). Notably, CD8^+^ T cell responses following immunization with GM3-αGC-OVA nanoparticles were of higher magnitude than responses observed after immunization with GM3-OVA liposomes in combination with αCD40/poly I:C, a potent adjuvant we have previously used [9]. This underlines the strength of including αGC as adjuvant in our vaccination strategy. Moreover, upon re-stimulation with MHC I-restricted OVA 257–264 peptide (SIINFEKL), the splenocytes from GM3-αGC-OVA-injected mice displayed the largest proportion of IFNγ-producing CD8^+^ T cells, confirming the superior immune activating capacity of GM3-αGC-OVA nanoparticles. Immunization with GM3-αGC-OVA liposomes also led to induction of IFNγ^+^ CD4^+^ T cells; however, the frequency was higher when GM3-OVA liposomes were co-injected with αCD40 and poly I:C.

The potential of GM3-αGC-OVA nanoparticles to stimulate antigen-specific B cell responses was also determined. Already on day 7 p.i., OVA-specific germinal center B cells were detected in the spleen. Interestingly, the inclusion of αGC in liposomal formulations led to an increase in the frequency of OVA specific B cells, while comparable frequencies were induced in mice injected with αGC-OVA and GM3-αGC-OVA formulations, indicating that addition of GM3 to αGC-OVA liposomes did not further increase B cell activation (Figure 2D). Total immunoglobulin (Ig) levels in nanoparticle immunized mice were in line with observed B cell frequencies. The GM3-OVA formulation, which lacks adjuvant capacity either in the form of liposome-incorporated αGC or co-injected αCD40/poly I:C, also lacked B cell-activating potential.

In summary, these data not only show that the inclusion of αGC leads to potent NKT and NK cell activation and therefore effective activation of innate players of the immune system, but also emphasize the superior capacity of GM3-αGC-OVA liposomes to stimulate CD8^+^ T cells, CD4^+^ T cells and B cells and therefore demonstrate the effectiveness of including αGC and GM3 in one vaccine formulation.

### 3.3. CD169^+^ Macrophages are Necessary for Induction of CD8^+^ T Cells Responses, but not for B Cell, NKT and NK Cell Activation Generated by GM3-αGC-OVA Liposomes

Since the major recipient of GM3-αGC-OVA liposomes, CD169^+^ macrophages, express CD1d (Appendix A), and considering the fact that this formulation induced potent NKT activation, we aimed to formally address the role of CD169^+^ macrophages in our liposome vaccination platform. To this end, we made use of the CD169-DTR mouse model, in which CD169-expressing cells were effectively depleted after a single dose of diphtheria toxin (DT) two days prior to liposome administration (Appendix A). Surprisingly, at 2 h p.i. with CD169-targeting GM3-αGC-OVA liposomes, no difference in the frequency of IFNγ-producing NKT or magnitude of IFNγ signal was observed between CD169^+^ macrophage-proficient and CD169^+^ macrophage-deficient mice (Figure 3A). Similarly, when we investigated NK cell activation in DT-treated or WT mice administered with GM3-αGC-OVA liposomes, we found comparable IFNγ levels based on frequency and gMFI (Figure 3B). These results show that CD169^+^ macrophages are not required for the activation of NKT and NK cells observed following treatment with a αGC-OVA or GM3-αGC-OVA nanovaccine in the dose tested here.

Next, we assessed the contribution of CD169^+^ macrophages to liposome-mediated stimulation of adaptive immunity 7 days p.i. Interestingly, antigen-specific CD8^+^ T cell responses were almost completely abolished in mice lacking CD169^+^ macrophages (Figure 3C), indicating that the induction of SIINFEKL^+^ CD8^+^ T cells after the injection of GM3-αGC-OVA liposomes is largely dependent on CD169^+^ macrophages. In contrast, the presence of CD169^+^ macrophages appeared to be irrelevant for activation of antigen-specific IFNγ-producing CD4^+^ T cells or B cell responses illustrated by both proportion of antigen-specific GC B cells and production of anti-OVA Ig (Figure 3D). This indicates that CD169^+^ macrophages are dispensable for T-helper (Th) and B cell activation in our vaccination system. 

Overall, it is evident that upon vaccination with GM3-αGC-OVA liposomes, CD169^+^ macrophages play a crucial role in the activation of antigen specific CD8^+^ T cells, but their contribution to the activation of B cells, NKT and NK cells is minor at best.

### 3.4. cDC1 Play an Essential Role in GM3-αGC-OVA Liposomes-Mediated Activation of CD8^+^ T Cells

Besides exhibiting superior CD8^+^ T cell (cross-)priming capacity as APCs, cDC1 express CD1d (Appendix A) and are also known for the induction of potent adaptive immune responses via NKT cell activation [27]. Accordingly, we investigated the importance of cDC1 in stimulation of NKT cells and the generation of antigen-specific immunity upon vaccination with GM3-αGC-OVA liposomes, using Batf3KO mice that lack cDC1 (Appendix A). While no change in the NKT cell frequency was detected between WT and Batf3KO animals at 2 h p.i. with αGC-containing formulations (data not shown), analysis of the IFNγ production by NKT cells revealed decreased levels in Batf3KO mice (Figure 4A). When we studied NK cell responses in mice exposed to GM3-αGC-OVA and αGC-OVA nanovaccines, we observed decreased NK cell activation in animals lacking cDC1, based on lower frequency of IFNγ-producing cells and lower expression per cell (Figure 4B). These results expose the contribution of cDC1 to NKT and NK cell activation following immunization with αGC-containing liposomes, but it must be highlighted that the residual levels of IFNγ-producing NKT and NK cells are still substantial in absence of cDC1 suggesting that cDC1 is not solely responsible for this effect.

Previous reports have already shown the indispensability of cDC1 in the cross-priming of CD8^+^ T cells [42,43]. As expected, 7 days p.i. with all nanovaccines, no SIINFEKL^+^ CD8^+^ T cells were observed in Batf3KO animals, which was in sharp contrast to the relatively abundant (12%) antigen-specific CD8^+^ T cells detected in WT counterparts (Figure 4C). Similarly, the frequency of IFNγ-secreting CD4^+^ T cells was diminished in Batf3KO mice vaccinated with αGC-OVA and GM3-αGC-OVA liposomes, suggesting a partial role of cDC1 in activation of IFNγ-producing CD4^+^ T cell responses. Cross-presenting cDC1 did not appear to be strongly involved in the production of antigen specific immunoglobulins, while the generation of antigen-specific B cell responses was significantly decreased in Batf3KO mice after immunization with GM3-αGC-OVA liposomes (Figure 4D). 

In summary, these findings clearly demonstrate that while cDC1 is crucial for generation of antigen-specific CD8^+^ T cells in response to immunization with GM3-αGC-OVA liposomal formulations, their function in NKT, NK, CD4^+^ T and B cell activation is evident but less striking.

### 3.5. Immunization with GM3-αGC-OVA Liposomes Provides a Maturation Signal for DCs and Induces IL-12 Secretion

Since we observed a synergistic role of αGC and CD169 targeting in the induction of robust CD8^+^ T cell responses, but no enhanced activation of NKT upon exposure to GM3-αGC liposomes, we asked whether immunization with GM3-αGC liposomes would lead to enhanced DC maturation. Therefore, we evaluated NKT cell responses and DC activation following 16 h in vivo challenge with all liposomal formulations. In line with previous reports demonstrating that NKT can provide a maturation signal to DCs by the upregulation of CD40L and cytokine secretion [44], we detected clear NKT activation as reflected by high percentages of IFNγ and IL-4 secreting cells and upregulation of CD40L (Figure 5A). However, no differences were detected between the two αGC formulations. Similarly, both GM3-αGC-OVA and αGC-OVA liposome induced a comparably high proportion of IFNγ-producing NK cells (Appendix A). Secretion of NKT-derived cytokines coincided with increased expression of the maturation markers CD40, CD80 and CD86 on cDC1 and cDC2. CD70 was only upregulated on cDC2 and MHC class II was unchanged in both DC subsets (Figure 5B,C and Appendix A). Notably, co-injection with the adjuvant combination of αCD40/poly I:C also endowed GM3 liposomes with potent DC-maturing capacity. While our αGC-containing nanovaccines exhibited strong DC-activating properties, confirming the potency of αGC as an adjuvant, the addition of GM3 as CD169 targeting ligand did not further enhance the maturation signal to DC. In addition to the phenotypic analysis of cells, we also evaluated cytokines in the serum at 16 h p.i. Interestingly, we detected significantly increased levels of IL-12 after vaccination with the GM3-αGC nanovaccine and not with the αGC nanovaccine (Figure 5D), while no differences were observed in IFNy, IL-6, IL-10, CXCL10 or CCL5 levels (data not shown). Since IL-12 has been shown to be a master regulator of NKT-mediated anti-tumor responses [45,46], this cytokine may be involved in the superior immune activation observed in GM3-αGC liposome-immunized mice. Overall, it is clear that the inclusion of αGC acts as a strong adjuvant, but that the additional incorporation of GM3 does lead to higher IL-12 production.

## 4. Discussion

The potential of therapeutic vaccination in the context of cancer is marked by the emergence of DC-based modalities and multiple efforts are made to develop a vaccination strategy that could work in synergy with other immunotherapies such as ICI [47]. An effective combination treatment should reinvigorate and facilitate the priming of tumor-reactive CD8^+^ T cells [48]. This can be achieved by the use of a vaccine that leads to optimal antigen presentation by DCs and concomitantly activates these DCs by the inclusion of an adjuvant. Here, we analyzed the immune activating capacity of systemically delivered GM3-αGC liposomes to enhance CD8^+^ T cell responses through additional NKT cell activation and investigated the requirement of CD169^+^ macrophages and cDC1s for its efficacy.

Our studies clearly show that the novel combined liposomal formulation of GM3 and αGC is superior in the activation of antigen specific CD8^+^ T cell and B cell responses, when compared to liposomes formulated with the single components. Interestingly, the GM3-αGC formulation outperformed immunization with GM3 liposomes co-injected with the potent adjuvant aCD40/poly I:C with regard to stimulation of CD8^+^ T cells and performed equally well in inducing B cell responses, while CD4^+^ T cell responses were only present with the aCD40/poly I:C adjuvant. This indicates that we have designed a very efficient vaccination platform that induces robust CTLs as well as B cell responses. However, future studies will be necessary to determine the prophylactic and therapeutic anti-tumor potential of GM3- αGC liposome vaccination in a tumor model system.

In the setting of systemic liposomal delivery, the addition of αGC and subsequent NKT activation serves as a substitute for CD4^+^ T cell help and results in the licensing of DC [27,49,50,51]. Indeed, our αGC-containing formulations exhibited strong adjuvant properties reflected by NKT cell activation and subsequent DC maturation, but we did not observe a further increase upon GM3 incorporation. However, we did detect increased levels of IL-12 in the serum of GM3-αGC immunized mice. IL-12 has been recognized as pivotal cytokine in CD8^+^ T cell proliferation and as potent inducer of anti-tumor immunity [47,52]. Langerin-expressing CD8α^+^ DC that reside in the MZ have been shown to produce IL-12 after systemic αGC delivery [53,54]. Additionally, cDC1-derived IL-12 was shown to be required for induction of Th1 [55]. This suggests that the GM3 inclusion in αGC liposomes may enhance the IL-12 production by NKT-activated cDC1, but the exact mechanism still needs to be elucidated.

Next to the interaction between NKT cells with cDC1, αGC-loaded SCS CD169^+^ macrophages have been shown to establish long-lasting interactions with NKT [22]. The specific targeting of αGC to splenic CD169^+^ macrophages with a synthetic CD169 ligand efficiently stimulated NKT cell responses [20]. This indicates that NKT cells can interact with αGC-presenting cDC1 as well as CD169^+^ macrophages. Interestingly, a few studies have demonstrated that CD1d molecules also present GM3 and that this can lead to NKT cell activation [56,57]. The combination of GM3 and αGC could potentially lead to a qualitatively different NKT cell activation. However, we did not detect NKT cell differences with regard to surface activation markers and cytokine expression and further research will be necessary to elucidate the synergism of GM3 and αGC incorporation in liposomes for CD8^+^ T cell activation.

Since the GM3-αGC liposome formulation was so efficient in stimulating CD8^+^ T cell responses, we performed mechanistic studies to determine the contribution of CD169^+^ macrophages and cDC1 for T cell, B cell, NKT and NK cell priming. Our studies clearly demonstrate that CD169^+^ macrophages and cDC1 are both necessary for the induction of CD8^+^ T cell responses after the systemic delivery of GM3-αGC liposomes. In the absence of either of these cell subsets, antigen-specific CD8^+^ T cell responses were completely abrogated. A similar observation has been made when CD169^+^ macrophages were targeted with CD169-OVA conjugated antibodies [11,14]. Apparently, the CD169^+^ macrophage subset collaborates with cDC1 to effectively prime CD8^+^ T cells. Although the exact mechanism of such collaboration is yet to be revealed, it is tempting to attribute it to antigen transfer to cDC1, as reported earlier [9,11], or to as yet unknown mechanisms in which CD169^+^ macrophages provide a platform for NKT and cDC1 interaction. Future studies are required to reveal the exact mechanism of this complex interplay.

Surprisingly, neither GM3 modification of αGC-containing liposomes nor DT-induced depletion of CD169^+^ macrophages affected NKT activation, as reflected by similar levels of IFNγ production. This could potentially be attributed to the relatively high dose of αGC in our liposomes, which already assured sufficient delivery of αGC to CD1d-expressing cells in the spleen and therefore hindered the detection of increased NKT activation after the delivery of GM3-aGC modified liposomes. This hypothesis is supported by other reports which have shown αGC to be effective in triggering of robust T cell responses upon i.v. administration of formulations encapsulating as little as 2–5 ng of the glycolipid [35,58]. Alternatively, we might be revealing a redundancy in the immune system where other CD1d-expressing cell types could take over NKT cell activation in the absence of CD169^+^ macrophages. The same redundancy might be responsible for the small, but significant, reductions in NKT and NK cell responses in Batf3KO mice lacking cDC1. A previous study showed that in the absence of Langerin^+^ CD8α^+^ DC no IL-12 could be detected, while IL-4 production by NKT was still present; however, subsequent IL-12-dependent NK cell production of IFNγ was also reduced [49]. This is in line with our data, as we observed a stronger decrease in NK cell responses than in NKT responses in Batf3KO mice. Barral et al. showed that CD11c ^+^ DCs, as well as MZ macrophages, contributed to NKT activation upon i.v. immunization with αGC-containing particles. Upon depletion of these cell subsets with clodronate liposomes, the authors observed a remaining 23% IFNγ ^+^ NKT, indicating the involvement of other CD1d-expressing cell types [22]. Indeed, CD169^+^ macrophages, cDCs as well as B cells express CD1d, and thus are equipped to directly activate NKT (Appendix A) [59,60,61]. Therefore, it remains to be determined which cell(s), besides CD169^+^ macrophages and DCs, play an additional role in the stimulation of NKT cells upon immunization with GM3-αGC liposomes.

We are convinced that our nanovaccine containing a combination of GM3, αGC and antigen offers tremendous potential for clinical applications. First of all, pre-clinical testing of nanoparticle-encapsulated αGC in combination with antigen has already been shown to result in delayed tumor growth and/or prolonged survival in B16 melanoma models [30,33,34,35]. Additionally, αGC has been evaluated in a multitude of clinical trials, either in a free form, encapsulated form or via ex vivo loaded DC, and has been proven safe [62,63,64]. Unfortunately, so far, clinical outcomes showed only minimal effect on overall survival (OS) [62]. Although multiple factors are involved in the limited effect on OS, such as the immunosuppressive environment of the tumor and general clinical complexity, there is a clear rationale for the improvement of αGC-based therapies. We postulate that as a combination treatment modality, nanovaccines containing GM3 and αGC could have a better potential as an immune-stimulating agent. Moreover, owing to their flexibility in formulation, low immunogenicity and biodegradability, liposomes are a popular and effective vehicle for delivery of anti-cancer drugs, with many FDA-approved formulations reaching the clinic [65]. More recently nanoparticle-based strategies have also been tested for the delivery of immunostimulatory agents with promising results/clinical benefit (e.g., RNA lipoplexes) [66,67].

## 5. Conclusions

To conclude, in this study we designed a liposome-based vaccination platform containing antigen combined with GM3 and αGC that offers evident potential for further development and evaluation in pre-clinical studies as an anti-tumor vaccination strategy.

## Figures and Tables

**Figure 1 vaccines-09-00056-f001:**
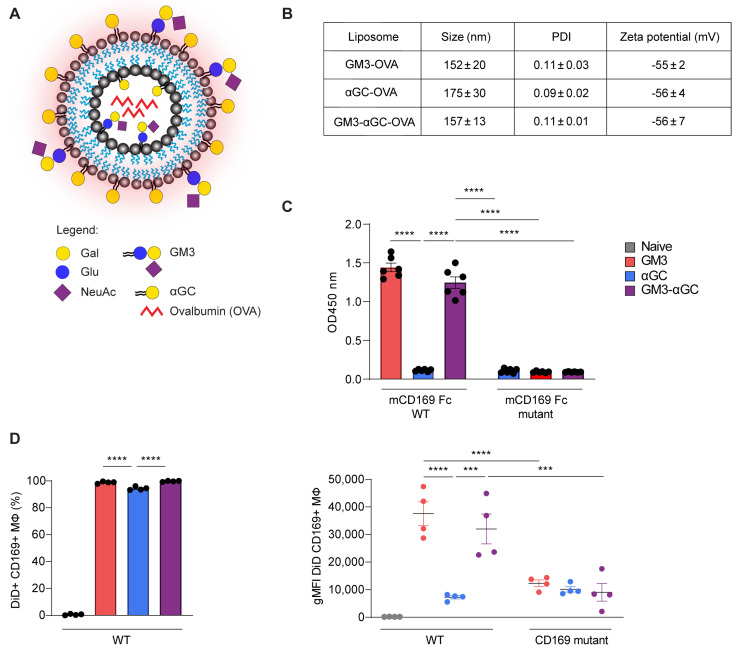
Liposome characterizations and their targeting potential to CD169+ macrophages. (**A**) Schematic representation of OVA-containing, DiD-labelled (red glow) nanoparticles with αGC and/or GM3 incorporated into the bilayer. Gal, galactose; Glu, glucose; NeuAc, sialic acid; GM3, ganglioside GM3; αGC, alpha-galactosylceramide. (**B**) Liposomal size, PDI (polydispersity index) and zeta potential. The data are the mean ± SEM of three liposome batches (**C**) Liposome binding to mouse CD169-Fc WT or CD169-Fc mutant complexes determined by ELISA. The data are the mean ± SEM from three liposome batches measured in duplicate. (**D**) Liposome uptake at 2 h post immunization (p.i.) by CD169+ macrophages from WT mice illustrated by the frequency of DiD+ cells (left panel) and the DiD fluorescence signal indicated as geometric mean fluorescence intensity (gMFI) from WT and CD169 mutant mice (right panel), determined by flow cytometry. The data are the mean ± SEM from one experiment with four mice per group. Each symbol represents one mouse (one-way ANOVA with Tukey’s test: *** *p* < 0.001, **** *p* < 0.0001).

**Figure 2 vaccines-09-00056-f002:**
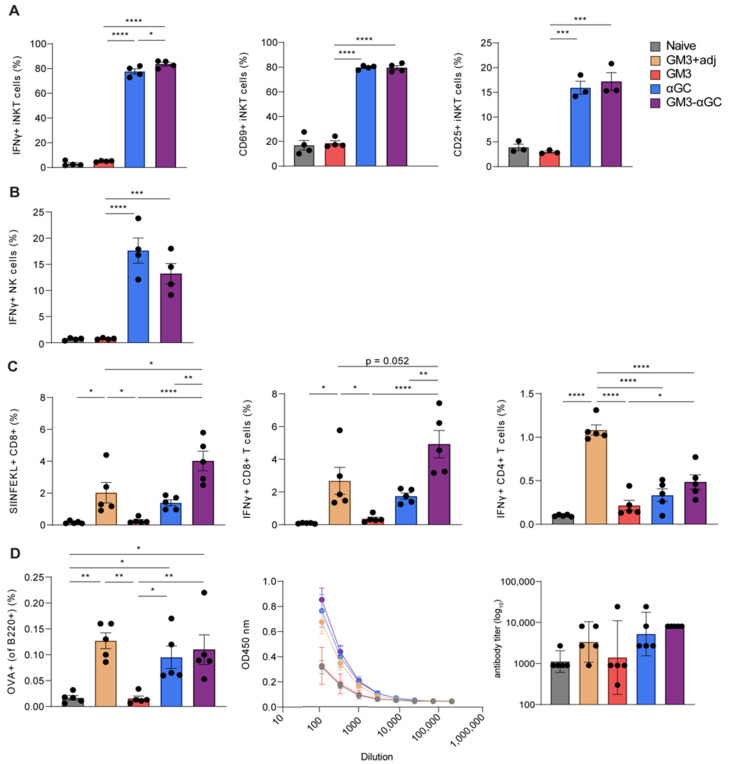
GM3-αGC liposomes activate natural killer T-cells (NKT) and natural killer (NK) cells and induce robust CD8^+^ T cell and B cell responses. GM3+adj represents GM3 liposome co-injected i.v. with αCD40/poly I:C. The activation of (**A**) NKT and (**B**) NK cells illustrated by frequency of CD25^+^, CD69^+^ and IFNγ-producing cells at 2 h p.i. with liposomes, determined by flow cytometry (**C**,**D**) The activation of antigen-specific T and B cells illustrated by frequency of MHC I-restricted OVA 257–264 peptide (SIINFEKL)^+^ CD8^+^ T cells, IFNγ production by CD8^+^ and CD4^+^ T cells and the frequency of OVA^+^ germinal center B cells (**D**, left panel) determined by flow cytometry at day 7 p.i. with liposomes. (**D**, middle and right panels) Serum anti-OVA Ig titer determined by ELISA on day 7 p.i. The data are the mean ± SEM (**A**–**D**) or geometric mean + 95% CI (**D,** right panel) from one experiment with 3–5 mice per group. Each symbol represents one mouse (one-way ANOVA with Tukey’s test: * *p* < 0.05, ** *p* < 0.01, *** *p* < 0.001, **** *p* < 0.0001).

**Figure 3 vaccines-09-00056-f003:**
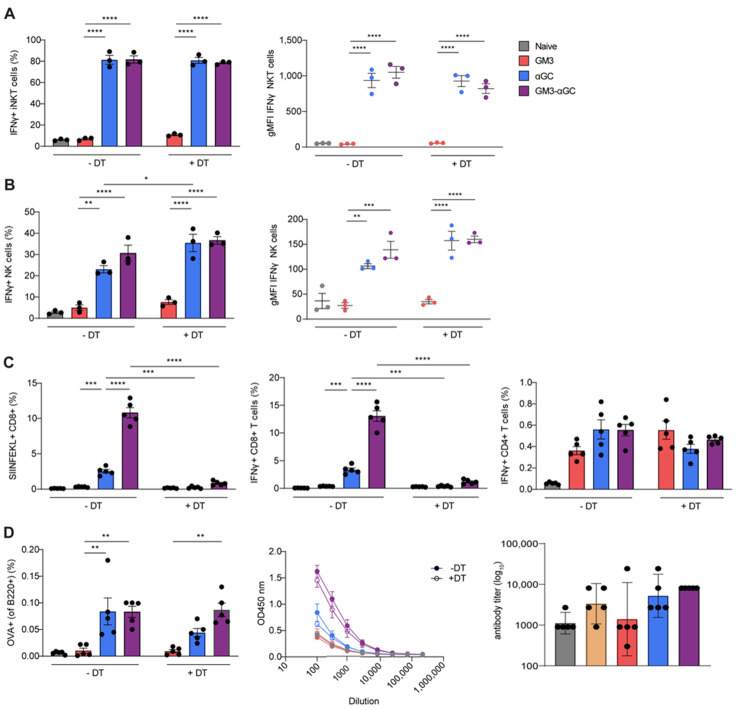
Upon challenge with GM3-αGC liposomes CD169^+^ macrophages are indispensable for the activation of OVA-specific CD8^+^ T cells but not for activation of NKT, NK and B cells. The activation of (**A**) NKT and (**B**) NK cells in CD169-DTR animals treated with DT or left untreated, analyzed at 2 h p.i. with liposomes illustrated by the frequency of the IFNγ-producing cells and gMFI, determined by flow cytometry. (**C**,**D**, left panel) The activation of antigen-specific T and B cells in CD169-DTR animals treated with DT or left untreated, analyzed at day 7 p.i. with liposomes illustrated by frequency of SIINFEKL+ CD8+ T cells, IFNγ production by CD8^+^ and CD4^+^ T cells and frequency of OVA+ germinal center B cells, determined by flow cytometry. (**D**, middle and right panels) The activation of B cells in CD169-DTR animals treated with DT (closed symbols) or left untreated (open symbols) on day 7 p.i. with liposomes illustrated by serum total anti-OVA Ig titer, determined by ELISA. The data are the mean ± SEM (**A**–**D**) or geometric mean + 95% CI (**D**, right panel) from one experiment with 3–5 mice per group. Each symbol represents one mouse (one-way ANOVA with Tukey’s test: * *p* < 0.05, ** *p* < 0.01, *** *p* < 0.001, **** *p* < 0.0001).

**Figure 4 vaccines-09-00056-f004:**
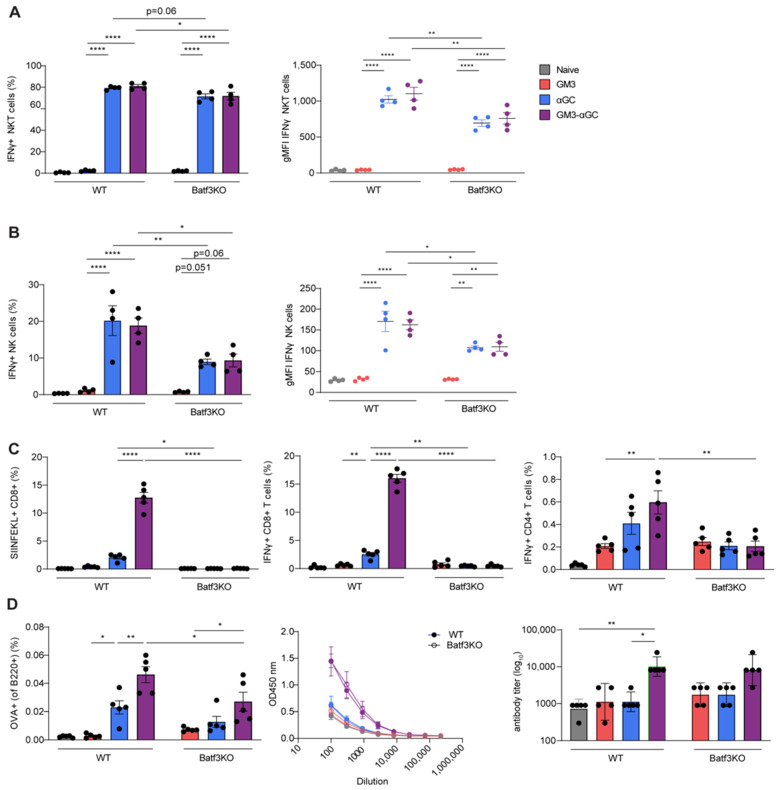
Vaccination with GM3-αGC liposomes triggers NKT, NK cell and CD8^+^ T cell responses that are dependent on cDC1.Activation of (**A**) NKT and (**B**) NK cells in WT and Batf3KO animals analyzed at 2 h p.i. with liposomes, illustrated by frequency of IFNγ-producing cells and gMFI, determined by flow cytometry. (**C**,**D**, left panel) Activation of antigen-specific T and B cells in WT and Batf3KO animals analyzed at day 7 p.i. with liposomes illustrated by the frequency of SIINFEKL^+^ CD8+ T cells, IFNγ production by CD8^+^ and CD4^+^ T cells and the frequency of OVA^+^ germinal center B cells, determined by flow cytometry (**D**, middle and right panels). Activation of B cells in WT (close symbols) and Batf3KO (open symbols) animals on day 7 p.i. with liposomes illustrated by serum total anti-OVA Ig titer, determined by ELISA. The data are the mean ± SEM (**A**–**D**) or geometric mean + 95% CI (**D**, right panel) from one experiment with 3–5 mice per group. Each symbol represents one mouse (one-way ANOVA with Tukey’s test: * *p* < 0.05, ** *p* < 0.01, **** *p* < 0.0001).

**Figure 5 vaccines-09-00056-f005:**
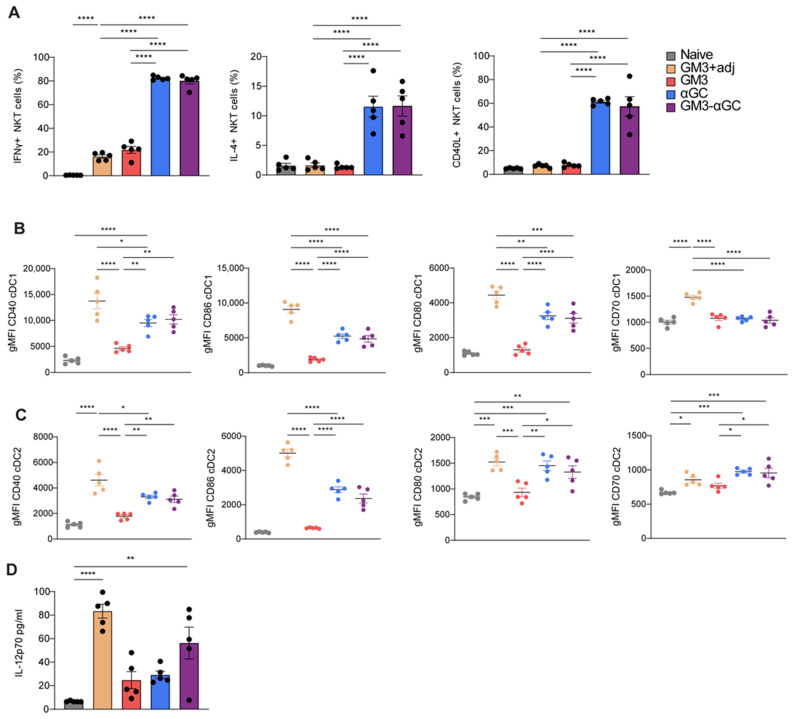
GM3-αGC liposome-activated NKT cells provide maturation signals to DCs and subsequently induce IL-12 production. GM3+adj, GM3 liposome co-injected i.v. with αCD40/poly I:C. (**A**) The activation of spleen NKT cells analyzed at 16 h p.i. with liposomes, illustrated by the frequency of IFNγ and IL-4 producing CD1d PBS57^+^ NKT cells and CD40L expression, determined by flow cytometry. (**B**,**C**) The expression of CD40, CD86, CD80 and CD70 by cDC1 (**B**) and cDC2 (**C**) illustrated by gMFI determined by flow cytometry. (**D**) IL-12 levels measured in the serum at 16 h p.i. with liposomes. The data are mean ± SEM from one experiment with 3–5 mice per group. Each symbol represents one mouse (one-way ANOVA with Tukey’s test: * *p* < 0.05, ** *p* < 0.01, *** *p* < 0.001, **** *p* < 0.0001).

**Table 1 vaccines-09-00056-t001:** The antibodies and fluorescently labelled reagents used for flow cytometry.

Antigen/reagent	Clone	Fluorochrome	Company
NK1.1	PK136	BV711	Biolegend
CD25	PC61	BV650	Biolegend
CD69	H1.2F3	AF700	Biologend
CD3	KT-3	Alexa Fluor 488	In-house made
CD4	GK1.5	BV510	Biolegend
CD8	53-6.7	PerCP-Cy5.5	Biolegend
IL-4	11B11	BV421	Biolegend
CD169	SER-4	Alexa Fluor 488	In-house made
B220	6B2	Alexa Fluor 405	In-house made
F4/80	T45-2342	PE-CF594	BD Biosciences
CD8a	53-6.7	PE-Cy7	BD Biosciences
CD11c	HL3	BV650	BD Biosciences
I-A/I-E	M5/114.15.2	PE	eBioscience
I-A/I-E	M5/114.15.2	Alexa Fluor 488	In-house made
CD80	16-10A1	PE	Immunotools
CD86	GL-1	PE-Cy7	BD Biosciences
XCR1	ZET	BV421	Biolegend
CD40	1C10	Biotin	In-house made
CD40L	MR1	PE-Cy7	Biolegend
CD70	FR70	biotin	BD Biosciences
CD8a	53-6.7	APC	BD Biosciences
CD44	KM81	FITC	Immunotools
H-K^b^/SIINFEKL	N/A	PE tetramer	LUMC, Leiden
B220	RA3-6B2	BV510	Biolegend
CD38	90/CD38	PE	BD Biosciences
GL7	GL-7	PE-Cy7	Biolegend
OVA	N/A	Alexa Fluor 488	Invitrogen
CD11a	M17/4	FITC	eBioscience
CD8a	53-6.7	PE-Cy7	BD Biosciences
CD4	GK1.5	PE	eBioscience
CD1d PBS-57	N/A	PE	NIH tetramer core facility
IFNγ	XMG1.2	APC	eBioscience

## Data Availability

Not applicable.

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
