# Peer review of "Liposomal Nanovaccine Containing α-Galactosylceramide and Ganglioside GM3 Stimulates Robust CD8+ T Cell Responses via CD169+ Macrophages and cDC1"

_vaccines, 2021, doi:10.3390/vaccines9010056_

Round 1
Reviewer 1 Report
- I think this is a nice manuscript.This a well conducted study and the authors further extend their recent finding of CD169 macrophages targeting vaccine using ganglioside liposomes (PNAS, 2020).
- It would be nice if the authors could provide a few murine xenograft data using murine cancer cell lines (such as B16-OVA or MC38-OVA etc.) with immune competent syngeneic mouse to fully demonstrate the better anti-cancer effect of this improved vaccine.
Author Response
Reviewer 1
- I think this is a nice manuscript.This a well conducted study and the authors further extend their recent finding of CD169 macrophages targeting vaccine using ganglioside liposomes (PNAS, 2020).
- It would be nice if the authors could provide a few murine xenograft data using murine cancer cell lines (such as B16-OVA or MC38-OVA etc.) with immune competent syngeneic mouse to fully demonstrate the better anti-cancer effect of this improved vaccine.
We thank the reviewer for his or her relevant comments. We agree with the reviewer that it would be very valuable to test the direct anti-tumor effect of our liposomal formulation containing GM3 and αGC. However, this would require tumor experiments of several months and we feel this is beyond the scope of our paper in which we show proof of concept data of enhanced immune responses by combination of GM3 and αGC in liposomes. To clarify that we agree with this suggestion we added in line 418-419 a statement that future studies should investigate the anti-tumor potential of our vaccination strategy and highlighted this in text with yellow.
Reviewer 2 Report
This is a well-performed study that is worth to be published in Vacines. The authors ask whether they can design a liposomal nanovaccine that uses the participation of both CD169+ macrophages and cD1 for full immunization including CD8+ T cell activation. The model is a mouse model and ovalbumine as antigen. The results are clear.
I propose some suggestions in order to improve the manuscript.
Line 60. The authors can consider to make a more in depth elaboration of CD16+ macrophages in the context of tumor-associated myeloid cells. What is written is correct, but really minimal. Clarifying more the importance of the CD16+ macrophages makes more stress on the importance of the study.
Line 94. "further". Compared to what ?, to aGC alone ? In the whole study, I miss a bit the starting point, and the improvement realized. Here it looks as the aGC alone is the basis, and the addition of GM3 is the novel element. Also this impression is created in line 426-427. However the figures are a bit designed in another way starting at the left side mostly with GM3. I would suggest to make the question a bit sharper, and then to see whether the design of figures match with this question at best.
Line 108. "CD169-DTR" mice. At this stage in the paper, we need a clarification of this term.
Line 208. "with CD169+ macrophages". In the introduction one speaks about MZ spleen macrophages and SCS lymph node macrophages. However, further in the experiments, I guess we speak only about spleen macrophages ? This should be somewhere clarified.
Legends to figures. Most figures contain one experiment with x mice per group. Data are then whown with dots, SD, SEM and sometimes CI95%. First of all, I do not see why sometimes SEM and sometimes CI95% is used. Second, most mouse groups contain 4-5 mice, not proving the existence of a normal distribution. I think that in all figures, we need the dots of the individual mice only, eventually with a median as line. I know that Anova is designed for normally distributed data, but I still believe the dots alone is the best way to present the individual data.
In the same line, having several groups with each time only 4-5 mice, it is quite hard to use Anova with multiple comparison correction. The data can be looked as obvious as such even without statistical analysis with very small groups within one experiment. Maybe somewhere one should shortly discuss this.
Figure 5D. Why do we not have the condition GM3+Adj ?
Line 479. "Therefore, ...". No. I do not agree. Indeed, the vaccines should be improved. But the lack of clinical efficacy and improvement of OS is not only at the side of the vaccine potency, but also due to the clinical complexity of most cancer diseases and the combination treatments of these patients. So more nuance should be used instead of "therefore".
Line 269-274. Layout should be as text.
Line 381-382. Two small type errors.
Author Response
Reviewer 2
This is a well-performed study that is worth to be published in Vacines. The authors ask whether they can design a liposomal nanovaccine that uses the participation of both CD169+ macrophages and cD1 for full immunization including CD8+ T cell activation. The model is a mouse model and ovalbumine as antigen. The results are clear.
We thank the reviewer for his or her valuable comments. Below we listed our responses to the comments raised by the reviewer.
Line 60. The authors can consider to make a more in depth elaboration of CD16+ macrophages in the context of tumor-associated myeloid cells. What is written is correct, but really minimal. Clarifying more the importance of the CD16+ macrophages makes more stress on the importance of the study.
In this manuscript we focus on targeting CD169+ macrophages in lymphoid organs, in particular splenic CD169+ macrophages and not so much tumor associated CD169+ macrophages. Therefore, we have chosen not to further elaborate on TAMs. The association of CD169+ macrophages with anti-tumor responses that we refer to in line 60 is further explained by examples in line 63-66.
Line 94. "further". Compared to what ?, to aGC alone ? In the whole study, I miss a bit the starting point, and the improvement realized. Here it looks as the aGC alone is the basis, and the addition of GM3 is the novel element. Also this impression is created in line 426-427. However the figures are a bit designed in another way starting at the left side mostly with GM3. I would suggest to make the question a bit sharper, and then to see whether the design of figures match with this question at best.
We agree that by including “further” the sentence asks for a comparison. We have replaced “further” with “could act synergistically to” in line 94 and highlighted this in yellow. Throughout the whole manuscript we compare the effect of CD169+ macrophage targeting alone (GM3) and iNKT activation alone (αGC) with the combination of both CD169+ macrophage targeting and iNKT activation together (GM3-αGC).
Line 108. "CD169-DTR" mice. At this stage in the paper, we need a clarification of this term.
We have added diphtheria toxin receptor before using the DTR abbreviation. This is changed in line 108 and highlighted in yellow.
Line 208. "with CD169+ macrophages". In the introduction one speaks about MZ spleen macrophages and SCS lymph node macrophages. However, further in the experiments, I guess we speak only about spleen macrophages ? This should be somewhere clarified.
The reviewer is correct that in this manuscript we target splenic CD169+ macrophages and not lymph node CD169+ macrophages. To better clarify this, we have added the word “splenic” into line 195 and 208 (highlighted in yellow).
Legends to figures. Most figures contain one experiment with x mice per group. Data are then whown with dots, SD, SEM and sometimes CI95%. First of all, I do not see why sometimes SEM and sometimes CI95% is used. Second, most mouse groups contain 4-5 mice, not proving the existence of a normal distribution. I think that in all figures, we need the dots of the individual mice only, eventually with a median as line. I know that Anova is designed for normally distributed data, but I still believe the dots alone is the best way to present the individual data.
In the same line, having several groups with each time only 4-5 mice, it is quite hard to use Anova with multiple comparison correction. The data can be looked as obvious as such even without statistical analysis with very small groups within one experiment. Maybe somewhere one should shortly discuss this.
Throughout the whole manuscript individual mice are represented by dots and the lines/bars represent mean and SEM. Although we use a relatively small sample size, we assume a normal distribution of our data and analysis with one-way ANOVA is a generally accepted method. We used Geometric mean + CI 95% only for the antibody titer data (Figures 2-4D only right panel), because of the log scale and discrete dilution steps of 1:3.
Figure 5D. Why do we not have the condition GM3+Adj ?
This is a valid point of the reviewer. We have included the data for GM3+Adj in the figure. We also noticed that incorrect absolute values for levels of IL-12 were accidentally plotted, because a 2X pre-dilution of serum was not yet taken into account. We have now included the correct values.
Line 479. "Therefore, ...". No. I do not agree. Indeed, the vaccines should be improved. But the lack of clinical efficacy and improvement of OS is not only at the side of the vaccine potency, but also due to the clinical complexity of most cancer diseases and the combination treatments of these patients. So more nuance should be used instead of "therefore".
The reviewer has a valid point by stating that the clinical complexity of cancer also plays part in effectivity of certain treatments. We have adjusted the sentence to add more nuance (line 481-482) and highlighted it.
Line 269-274. Layout should be as text.
We have changed the layout of line 269-273 and highlighted it yellow.
Line 381-382. Two small type errors.
Both type errors have been corrected and highlighted in yellow.